# Titanium Disulfide Based Saturable Absorber for Generating Passively Mode-Locked and Q-Switched Ultra-Fast Fiber Lasers

**DOI:** 10.3390/nano10101922

**Published:** 2020-09-26

**Authors:** Xinxin Shang, Linguang Guo, Huanian Zhang, Dengwang Li, Qingyang Yue

**Affiliations:** 1Shandong Provincial Engineering and Technical Center of Light Manipulations, Shandong Provincial Key Laboratory of Optics and Photonic Devices and Shandong Key Laboratory of Medical Physics and Image Processing, School of Physics and Electronics, Shandong Normal University, Jinan 250014, China; xinxin_shang10@163.com (X.S.); linguang_guo@163.com (L.G.); 2School of Physics and Optoelectronic Engineering, Shandong University of Technology, Zibo 255049, China

**Keywords:** High-damage 2D Titanium disulfide materials, ultra-fast optical modulation, mode-locked and Q-switched, saturable absorber

## Abstract

In our work, passively mode-locked and Q-switched Er-doped fiber lasers (EDFLs) based on titanium disulfide (TiS_2_) as a saturable absorber (SA) were generated successfully. Stable mode-locked pulses centred at 1531.69 nm with the minimum pulse width of 2.36 ps were obtained. By reducing the length of the laser cavity and optimizing the cavity loss, Q-switched operation with a maximum pulse energy of 67.2 nJ and a minimum pulse duration of 2.34 µs was also obtained. Its repetition rate monotonically increased from 13.17 kHz to 48.45 kHz with about a 35 kHz tuning range. Our experiment results fully indicate that TiS_2_ exhibits excellent nonlinear absorption performance and significant potential in acting as ultra-fast photonics devices.

## 1. Introduction

Due to the advantages of short pulse width, high peak power, simple structure, and excellent beam quality, pulsed fiber lasers have recently attracted significant attention because of their wide corresponding applications, such as medical treatment [1,2], fiber communications [3,4], environmental monitoring, and industry [5,6,7]. Passively mode-locked or Q-switched techniques are commonly-used efficient methods for generating ultra-fast pulsed lasers [8,9,10,11]. Passive techniques including nonlinear polarization rotation (NPR) [12], nonlinear amplifying loop mirror (NALM), and saturable absorber (SA) [13] are employed for demonstrating pulsed lasers operating from visible to mid-infrared optical band. Among them, SA plays an important role in demonstrating passively mode-locked and Q-switched lasers due to their unique polarization-independent properties. Previously, semiconductor saturable absorber mirrors (SESAMs), single-wall carbon nanotubes (SWCNTs), and graphene are three main fast SAs for achieving passively mode-locked and Q-switched laser operations [14,15,16,17,18]. Among them, SESAM exhibits repeatable absorption performance in acting as ultra-fast optical devices. However, it also possesses clear disadvantages such as high cost, narrow absorption band and low damage threshold [19,20]. SWCNT has the advantages of low cost, ultra-fast recovery time, a high damage threshold, and low saturation intensity. However, the bandwidth of SWCNT depends on its diameter, which leads to a large light scattering loss [21]. Graphene possesses a wide absorption spectrum band due to its Dirac-like electronic band structure [22,23], but its zero-bandgap property also limits its wide potential applications [24,25,26]. Recently, based on the development of layered materials, novel nanomaterials such as transition metal dichalcogenides (TMDs), black phosphorus (BP), MXene, and topological insulators (TIs) have emerged as ultra-fast optical devices and shown excellent absorption performance [27,28,29,30,31,32,33,34,35,36,37,38,39,40,41].

Recently, IV–VI group TMDs, which can be expressed with the chemical formula as MX_2_ (M = Mo, W, Ta, V, Nb, Re, Ti, etc; X = S, Se), have gained more attention in the field of ultrafast photonics due to their high third-order optical nonlinear susceptibility and suitable layer-independent bandgap value [42,43,44]. Titanium disulfide (TiS_2_), which is a novel IV–VI semiconductor with a layered structure, has a giant tunable band gap from ~0.05 eV for bulk structure to ~2.87 eV for 1–3 layers of structure [45,46]. The structural layers are coupled by a weak Van Der Waals force, offering the possibility for exfoliation from the bulk to several thin layers. Recently, TiS_2_ was reported as a superior nonlinear optical limiting material [47]. Additionally, TiS_2_ used as SAs for generating pulsed laser operations have also been investigated preliminarily [44,48,49]. The light from 1530 nm to 1565 nm is a conventional band (C-band) that represents the regular waveband. The optical fiber exhibits the lowest loss in the C-band and has a greater advantage in long-distance transmission systems. Applied in long-distance, ultra-long-distance, and submarine optical transmission systems, the C-band becomes more important. Therefore, 1.5-micrometer mode-locked fiber lasers should have more exploration and research. The advantages of stability, economy, and high damage threshold TiS_2_ prove that it is an excellent saturable absorber. The main purpose of our work is to explore the deep ultrafast photonics application of TiS_2_.

In this case, we demonstrated passively mode-locked and Q-switched Er-doped fiber lasers (EDFL) based on a TiS_2_-PVA film-type SA. Mode-locked laser centred at 1531.69 nm with a 3 dB bandwidth of 1.046 nm and clear Kelly sidebands was generated, which implied that the pulse width may be about 2.36 ps. Q-switched operation operated at a central wavelength of 1531.67 nm. The maximum pulse energy and minimum pulse duration were 12.7 nJ and 3.42 µs, respectively. By reducing the length of the laser cavity and optimizing the cavity loss, the maximum pulse energy and minimum pulse width were optimized to be 67.2 nJ and 2.34 µs, respectively.

## 2. Preparation and Characteristics of the TiS_2_ Material

In recent work, different methods have been adapted to prepare saturable absorbers, such as optically deposition, chemical vapor deposition (CVD), and deposition in tapered fibers. A saturable absorber with a sandwich structure prepared a PVA film was applied in this experiment. The saturable absorber prepared by this method is easy to control and transfer, and the preparation method is simple and the cost is low. The film-type TiS_2_-polyvinyl alcohol (PVA) SA used in our experiment was prepared by the method of the liquid-phase exfoliation and spin coating. Firstly, we added 0.3 g TiS_2_ powder into 30 mL of 30% alcohol for preparing TiS_2_ solution. The solution was placed in an ultrasonic cleaner for 6 h. Few-layer TiS_2_ nanosheets were obtained. Then, the TiS_2_ solution and 4 wt% PVA solution were mixed at a volume ratio of 1:1 and placed in the ultrasonic cleaner for 6 h for preparing uniform TiS_2_-PVA solution. Then, 50 μL dispersion solution was spin coated on a culture dish to form TiS_2_-PVA film, It was then put the culture dish into a drying oven at 35 °C for 12 h. Lastly, a piece of the film with the size of ~1 × 1 mm^2^ was gently cut off and attached on a clean FC/PC fiber ferrule as a proposed modulator.

For testing the surface morphology of the TiS_2_ nanosheets, the TiS_2_ powder was detected by a scanning electron microscope (SEM). Figure 1a depicts the SEM image of TiS_2_ nanosheets, as is shown. The TiS2 nanosheets possess a clear layered structure. The energy dispersion spectroscopy (EDS) spectrum of the TiS_2_ nanosheets is shown in Figure 1b. The corresponding peaks associated with titanium and sulfide are clearly observed. The atomic ratio of sulfur and titanium is 63:37, which corresponds to the formula of TiS_2_. Our results indicate that pure TiS_2_ nanosheets with a well layered-structure are prepared successfully.

The X-ray diffraction (XRD) of TiS_2_ was given in Figure 2a with characteristic peaks between 10° to 90° for characterizing its crystal structure. The strong diffraction peaks located at (001), (011), (012), and (110), which is consistent with the TiS_2_ crystal spectrum in the standard spectrum library (PDF-65-3372). In addition, Figure 2b shows the Raman spectrum of the TiS_2_ in the range of 50–400 cm^−1^ using the 532 nm excitation line at room temperature. Two typical Raman peaks at 230 and 332 cm^−1^, corresponding to the E_g_ and A_1g_ modes of the TiS_2_ are recorded, which shows a negligible shift in comparison with previous results [50,51,52].

The AFM image shown in Figure 3a was recorded to determine the thickness of TiS_2_ nanosheets by an atomic force microscope (AFM), and the corresponding height of the selected areas in Figure 3a were 12, 13, 13, and 16 nm, corresponding to 21–29 layers since the monolayer thickness of TiS_2_ is 0.57 nm. As sketched in Figure 3c, the transmission electron microscope (TEM) image indicates that the TiS_2_ nanosheets have a clear layered structure. Figure 3d presents the linear transmission spectrum of TiS_2_-PVA film on the substrate, and the substrate was also provided to show the linear absorption from TiS_2_ nanosheets. The dip near 400 nm in the transmission spectrum is a typical fingerprint of TiS_2_ nanosheets. The absorption band extends from 400 nm to 2000 nm. The nonlinear absorption property of the TiS_2_-PVA film was investigated by a power-dependent transmission technique, as shown in Figure 3e. The relationship between the transmittance and the optical intensity can be fitted by the function shown in the following Equation:$$T(I)=1-\Delta T.\exp(-I/I_{sat})-T_{ns}$$
where *T* and *T_ns_* denotes transmission and non-saturable loss, and Δ*T* is modulation depth, *I* and *I_sat_* are input intensity of laser and saturation intensity. The modulation depth and saturation intensity is calculated to be 13.19% and 17.97 MW/cm^2^, respectively. The measured results indicate that TiS_2_ is a promising broadband optical material for ultrashort pulse generation in lasers.

The saturable absorption theory can be explained by the Pauli blocking principle as a typical 2D material. Linear absorption can occur under low excitation intensity. The electrons distributed in the valence band can absorb the energy of the incident light and be excited into the conduction band when the energy of the incident light is greater than the bandgap value of the TiS_2_. Afterward, the hot electrons cool down almost immediately and cause the formation of a hot Fermi-Dirac distribution. In this case, the newly generated electron-hole pairs will prevent the initial potential inter-band optical transitions and photons absorption around the Fermi energy (−E/2). Lastly, due to the internal scattering of phonons, electrons and holes recombine and reach a state of equilibrium distribution. However, the concentration of photocarriers will increase instantaneously and the energy states near the edges of the conduction band and valence band will be filled under higher excitation intensity. Since no two electrons can reach the same state defined by the Pauli blocking principle, the absorption will be blocked. Therefore, specific frequency photons transmit the material without absorption. As is described, the bandgap value has a great significance for the design of ultra-fast photonics devices.

## 3. Experimental Setup

For testing the nonlinear absorption properties of the TiS_2_ SA, an all-fiber ring laser cavity was demonstrated and shown in Figure 4. A 28-cm long Er-doped fiber (LIEKKI Er80-8/125, nlight, Vancouver, WA, USA) with a group velocity dispersion of −22.6 ps^2^km^−1^ was used as the laser gain medium and pumped by a 980-nm laser diode (LD) with the maximum pump power of 1.3 W through a 980/1550 nm wavelength division multiplexer (WDM). Another port of the WDM was attached to the 90% port of an optical coupler (OC). A polarization-insensitive optical isolator (PI-ISO) was employed to ensure the unidirectional operation and two polarization controllers (PCs) were used to adjust the polarization state. In addition, a 50-m long single-mode fiber (SMF) with a dispersion parameter of ~17 ps/(nm × km) was added into the cavity to adjust the net dispersion. Furthermore, 10% output characteristics were analyzed by a 3-GHz photo-detector (PD 03) combined a digital oscilloscope (Wavesurfer 3054, Teledyne LeCroy, USA), a power meter (PM100D-S122C, Thorlabs, New Jersey, American),a radio-frequency spectrum (R&S FPC1000, Jena Germany), and an optical spectrum analyzer (AQ6317B, Yokogawa, Tokyo, Japan).

## 4. Results and Discussion

### 4.1. Mode-Locked Operation

Optimization before carrying out the experiment, the output characteristics of the fiber laser without inserting the TiS_2_ modulator into the ring laser cavity was tested. No mode-locked or Q-switched pulses were recorded by adjusting the pump power and polarization states of PCs. Then, the TiS_2_ modulator was inserted into the laser cavity, as sketched in Figure 4. When the pump power increased to 276 mW, stable mode-locked pulses were observed by adjusting the PCs carefully. The output characteristics are shown in Figure 5. The output power is 0.177 mW, which corresponds to a single pulse energy of 0.05 nJ. As shown in Figure 5a, a typical soliton spectrum with clear symmetric Kelly sidebands is obtained, which indicates that the laser operates at a traditional soliton mode-locked state. Due to low output power, it is difficult for us to measure the real characteristics of pulse width. The central wavelength located at 1531.69 nm with half wavelength full width (FWHM) of 1.046 nm, which implied that the pulse width of the laser is about 2.36 ps based on the soliton theory [53]. Figure 5b depicts the pulse train on oscilloscope. The interval of pulse train is 291.3 ns, which matches the round-trip time of the cavity. Figure 5c presents the radio frequency (RF) spectrum located at the fundamental repetition rate of 3.43 MHz with a signal-to-noise ratio of ~60 dB, corresponding to the cavity length of 59.8 m. The higher harmonics within a wide bandwidth of 34 MHz is shown in Figure 5d for investigating the long-term stability. The interval of the repetition rate is 3.43 MHz. The RF results indicate that stable mode-locked pulse operation is achieved.

### 4.2. Q-Switched Operation

Then, for obtaining large-energy Q-switched operating commonly obtained within short-length laser cavity, the length of the ring laser cavity was adjusted by cutting off 30 m SMF. Stable self-started Q-switched operation was observed under the pump power of 221 mW. The recorded pulse train with good stability is shown in Figure 6a. The pulse-to-pulse time is 60.24 μs, which corresponds to a pulse repetition rate of 16.6 kHz (the RF spectrum is shown in Figure 6d). The single pulse profile is presented in Figure 6b. The FWHM pulse duration is about 7.27 µs. The emission spectrum is provided in Figure 6c. The central wavelength is about 1531.67 nm.

The repetition rate and pulse duration characteristics under different pump powers are described in Figure 7a. As sketched in Figure 7a, the pulse duration reduces from 7.27 to 3.42 µs and the repetition rate increases from 16.6 to 26.7 kHz with the increase of the pump power. Figure 7b describes the output power and pulse energy as a function of pump power. The maximum pulse energy is 12.7 nJ under the pump power of 407 mW. The pulse energy and pump power have a linear relationship under the pump power of 400 mW, but the pulse energy was limited when the pump power exceeded 400 mW. The measured minimum pulse width was 3.42 µs at a pump power of 374 mW. It is considered that the pulse duration could be shortened further by reducing the length of the laser cavity [54] and by optimizing the cavity loss [55,56].

For getting the shorter pulse duration and the more stable fiber laser, we continued to adjust the length of cavity to 7.9 m by cutting off the SMF. Stable self-started Q-switched operation was achieved at a threshold pump power of 79 mW. Figure 8a displays the Q-switched pulse train recorded by the oscilloscope. The interval of pulse train is 20.9 µs, which corresponds to the repetition rate of 48.8 kHz (the yellow peak in Figure 8c). The corresponding minimum pulse width is 2.3 µs (shown in Figure 8b) at the pump power of 352 mW. The repetition rate under different pump power is presented in Figure 8c. All of the repetition rates match well with the Q-switched pulse-to-pulse time and have high signal-to-noise ratios. The experimental results exhibit that high stable Q-switched pulses are obtained by reducing the length of the laser cavity to optimize the cavity loss. The corresponding optical spectrum under different pump power was provided in Figure 8d. The center optical spectrum was located at 1556 nm when the pump power was within 300 mW.

Figure 9a shows the pulse width and the repetition rate as a function of the pump power. By increasing the pump power from 79 mW to 352 mW, the repetition rate monotonically increases from 13.17 to 48.45 kHz with about a 35-kHz tuning range. The pulse width reduced from 8.49 to 2.34 µs with increasing pump power. This phenomenon is due to the gain compression in the Q-switched fiber laser [55]. Figure 9b presents the average output power and the single pulse energy under different pump powers. The maximum average output power and single pulse energy are 3.26 mW and 67.24 nJ, respectively. All the results prove that the cavity loss is reduced by shortening the length of the cavity.

In 2018, Zhu et al. reported the TiS_2_ used as SA for the first time, which is of great significance [44]. As shown in Table 1, we compared the parameters of the laser obtained by our experiment with the parameters of their laser. They used the optically deposition method to prepare SA while the sandwich structure was adopted in our experiment. The optical modulation depth of our SA was 13.19% higher than 8.3% in their experiment. In the mode-locked and Q-switched pulse operation, although the center wavelength obtained in this experiment has a blue shift for the same nanomaterial, this is our next step to explore the influence factors of the blue shift of the laser wavelength. They can get the pulse duration of 812 fs in a mode-locked pulse, and 2.36 ps pulse was obtained in our work. Higher single pulse energy was obtained in both a mode-locked and Q-switched operation than by Zhu, X. et al. [44]. The highest single pulse energy can reach 67.24 nJ. This shows that the fiber laser we designed has excellent characteristics.

## 5. Conclusions

To sum up, TiS_2_ nanosheets were prepared by the liquid-phase exfoliation method and its nonlinear optical properties were investigated experimentally. Based on TiS_2_-PVA film-type SA, both mode-locked and Q-switched Er-doped fiber lasers were achieved. Stable mode-locked pulse centred at 1531.69 nm with the pulse width of 2.36 ps was obtained. By reducing the length of the laser cavity and optimizing the cavity loss, self-started Q-switched operation with a maximum pulse energy and a minimum pulse width of 67.2 nJ and 2.34 µs was obtained. Our findings prove that TiS_2_ exhibits excellent absorption performance and will have wide applications in proposing ultrafast photonic devices.

## Figures and Tables

**Figure 1 nanomaterials-10-01922-f001:**
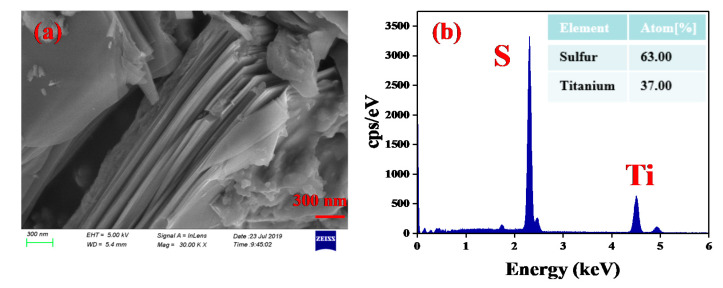
(**a**) SEM image of the TiS_2_ nanosheets. (**b**) EDS spectrum of the TiS_2_ nanosheets.

**Figure 2 nanomaterials-10-01922-f002:**
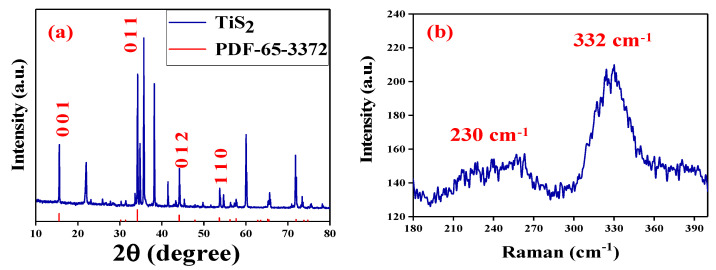
(**a**) XRD analyze for TiS_2_ nanosheets. (**b**) Raman spectrum of TiS_2_ nanosheets.

**Figure 3 nanomaterials-10-01922-f003:**
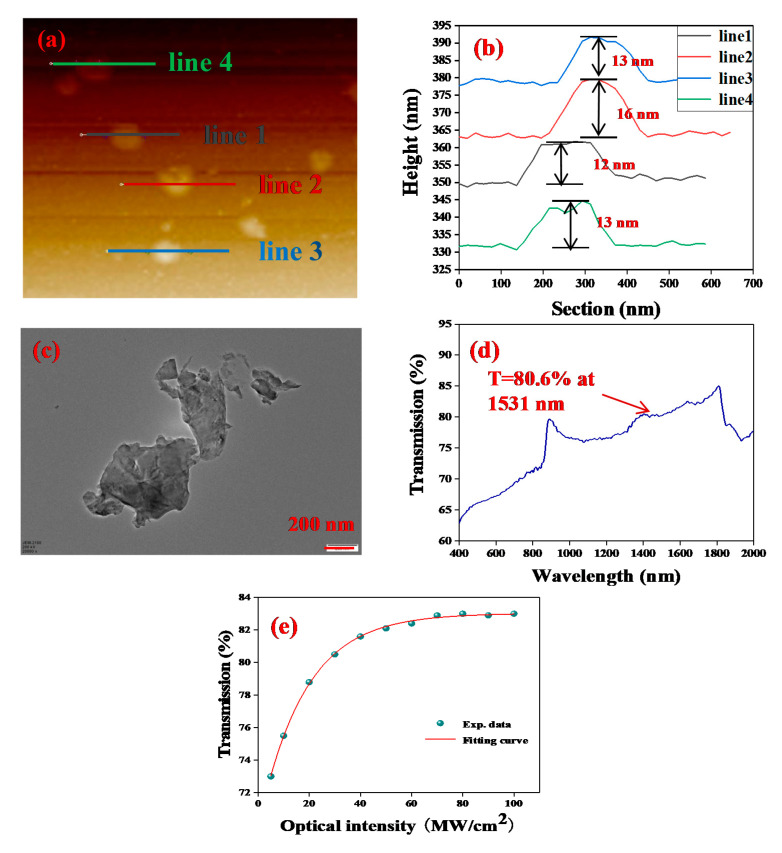
(**a**) AFM image of TiS_2_ nanosheets. (**b**) Corresponding height profile of the selected area in (**a**). (**c**) TEM image of TiS_2_ nanosheets. (**d**) Linear transmission of the TiS_2_-PVA film SA. (**e**) Nonlinear absorption property of the TiS_2_-PVA film.

**Figure 4 nanomaterials-10-01922-f004:**
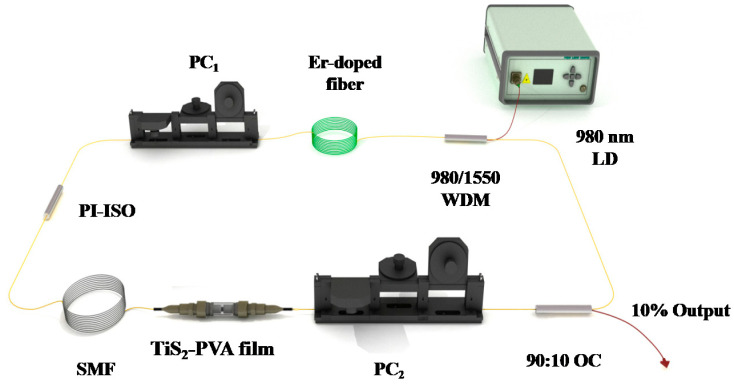
Schematic diagram of the experimental setup.

**Figure 5 nanomaterials-10-01922-f005:**
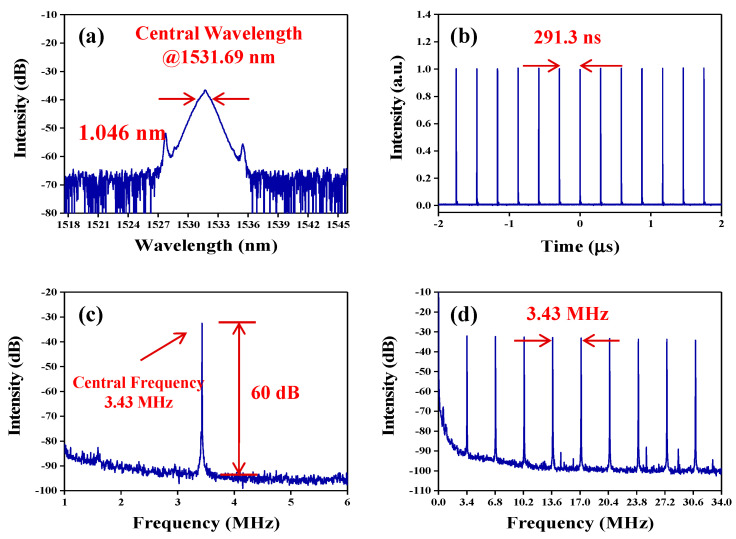
Typical mode-locked pulse characteristics at the pump power of 276 mW. (**a**) The output optical spectrum with clear Kelly sidebands. (**b**) The corresponding mode-locked pulse train. (**c**) The fundamental repetition rate. (**d**) The higher harmonics within a bandwidth of 34 MHz.

**Figure 6 nanomaterials-10-01922-f006:**
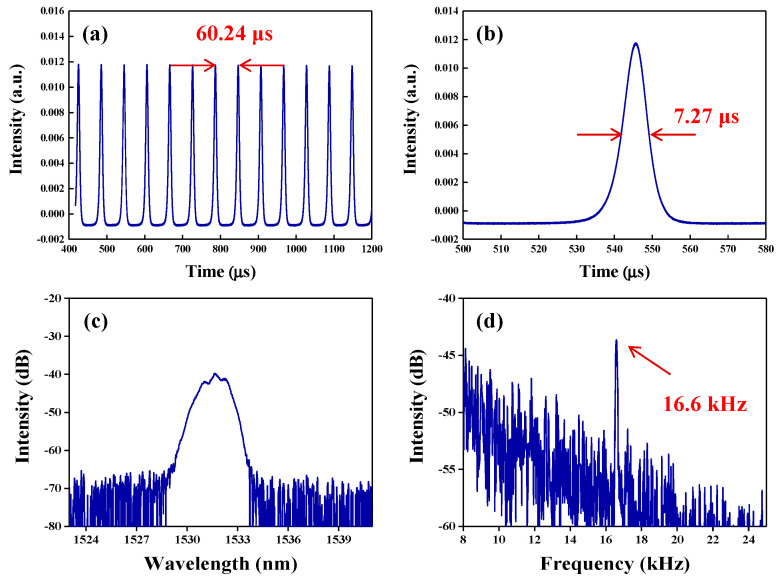
The Typical Q-switched pulse characteristics at the pump power of 221 mW. (**a**) Output pulse train at 16.6 kHz repetition rate. (**b**) Single pulse profile with 7.27 µs duration. (**c**) The emission spectrum. (**d**) The RF spectrum located at 16.6 kHz.

**Figure 7 nanomaterials-10-01922-f007:**
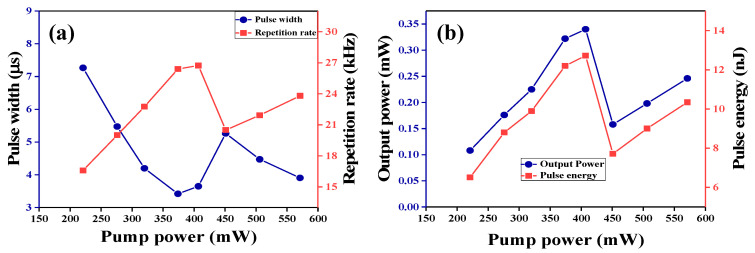
(**a**) Variation of repetition rate and pulse duration with pump power. (**b**) Output power and pulse energy as a function of pump power in the Q-switched fiber laser.

**Figure 8 nanomaterials-10-01922-f008:**
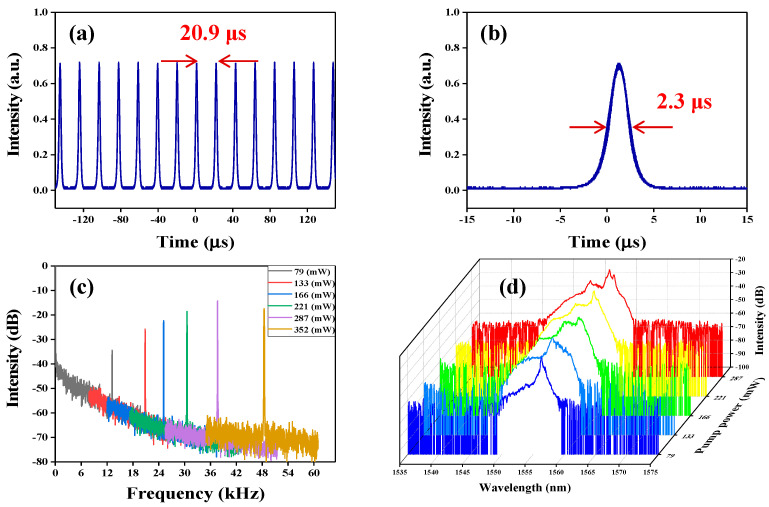
Typical Q-switched pulse characteristics with a 7.9-m long cavity. (**a**) Output pulse train at a 48.8-kHz repetition rate. (**b**) Single pulse profile with a 2.3-μs duration (**c**) and (**d**) the repetition rate under different pump powers and its corresponding emission spectrum.

**Figure 9 nanomaterials-10-01922-f009:**
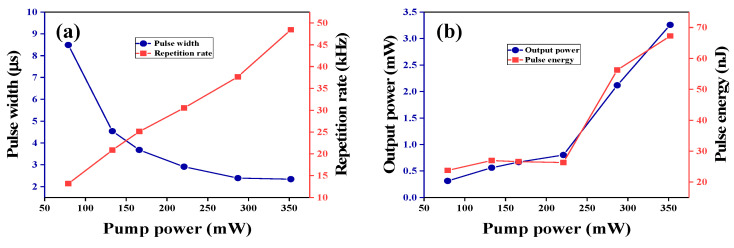
(**a**) Pulse width and repetition rate versus pump power. (**b**) Output power variation and pulse energy as a function of pump power in the Q-switched fiber laser.

**Table 1 nanomaterials-10-01922-t001:** A comparison of the characters of our fiber laser with [44].

Pulse Type	SA Fabrication	α_s_ (%)	*I*_sat_ (MW/cm^2^)	Wavelength (nm)	Pulse Duration	Fundamental Frequency (MHz)	Pulse Energy	Output Power (mW)	Ref.
Mode-locked	optically deposition	8.3	-	1563.3	812 fs	22.7	25.3 pJ	0.574	[44]
Q-switched				1560.2	4 µs	25.2 to 50.7 kHz	9.5 nJ	0.48	
Mode-locked	sandwich structure	13.19	17.97	1531.69	2.36 ps	3.43	0.05 nJ	0.177	Our work
Q-switched				1556	2.34 µs	13.17 to 48.45 kHz	67.24 nJ	3.26	

α_s_, modulation depth; *I*_sat_, saturable intensity;

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
