# Peer review of "Titanium Disulfide Based Saturable Absorber for Generating Passively Mode-Locked and Q-Switched Ultra-Fast Fiber Lasers"

_nanomaterials, 2020, doi:10.3390/nano10101922_

Round 1
Reviewer 1 Report
I have one major concern related to the paper. Namely, I struggle to see novelty in developing a modelocked fiber laser operating at 1.5 micrometers. This topic has been very profoundly explored during the last decade and many solutions to this problem have been proposed. In my view the authors should explain why their solution is better to what already has been shown. Otherwise it is just another laser and there are, as mentioned, many papers on this topic already published. Stating only that the laser is generating pulses at 1.5 micrometers, is in my view not sufficient to justify the publication. Also many materials were used succesfully as saturable absorbers. Why add another one? What does the suggested material improve?
Author Response
Response: Thank you for your valuable comment and suggestion. The light from 1530 nm to 1565 nm is conventional band (C-band) represents the regular waveband. The optical fiber exhibits the lowest loss in the C-band and has a greater advantage in long-distance transmission systems. Applied in long-distance, ultra-long-distance and submarine optical transmission systems, the C-band becomes more and more important. Therefore, 1.5 micronmeters mode-locked fiber lasers should have more exploration and research.
In this experiment, the mode-locked pulse and the Q-switched pulse are obtained in the same laser cavity. The pulse width of the mode-locked can reach 2.36 ps. In the Q-switched operation, we have tried different length of single-mode fiber, and the stability can still be obtained under different environments.
In recent years, two-dimensional materials have been widely used as saturable absorbers and have achieved very good results. Transition metal dichalcogenides is very suitable for saturable absorber due to its special microstructure and energy band structure, but for different transition metal dichalcogenides, there are some differences between them. Therefore, we can fully understand the application of transition metal dichalcogenides as a saturable absorber through the research on TiS2. The advantages of stability, economy, and high damage threshold TiS2 prove that it is an excellent saturable absorber. This helps us to explore TiS2's optoelectronic properties and have greater applications.
Correction is marked with red colour and underline in the manuscript and listed below.
In Line 55 Page 2
The light from 1530 nm to 1565 nm is conventional band (C-band) represents the regular waveband. The optical fiber exhibits the lowest loss in the C-band and has a greater advantage in long-distance transmission systems. Applied in long-distance, ultra-long-distance and submarine optical transmission systems, the C-band becomes more and more important. Therefore, 1.5 micronmeters mode-locked fiber lasers should have more exploration and research. The advantages of stability, economy, and high damage threshold TiS2 prove that it is an excellent saturable absorber.

Reviewer 2 Report
This paper reports the demonstration of mode-locking and passive Q-switching in an erbium-doped fiber laser with a saturable absorber made of a film of titanium disulfide (TiS2).
The results are sound, and the description of the experiment is accurate, especially regarding the characterization of the TiS2 material. Unfortunately, the results are very similar to those of Ref. [44]. The only difference is that while in [44] the cavity length was kept fixed and the transition from mode-locking to passive Q-switching is achieved by controlling the cavity gain and losses, here the transition is achieved mainly by varying the cavity length. I am afraid that for this reason the paper should be regarded as incremental and not suitable for publication on Nanomaterials.
The authors should revise the paper in such a way to emphasize the differences between their experiment and that of [44]. Perhaps the impact of the paper can be increased by adding a comparison between experimental results and theoretical predictions based on a model of laser with saturable absorber, a feature which is missing in [44]. Still on the theoretical side, I wonder why the saturation curve of Fig. 2(e) is fitted with an exponential function rather than with a two-level saturation function.
Another minor point is that the symbols and colors in Fig. 9(a) are reversed with respect to Fig. 7(a). Also it is better to represent the data in Fig. 7(a) as functions of the Pump power instead of Output power, again for consistency with other figures.
The English is good but strangely the verb “to process” is used with the meaning of “to possess”.
Author Response
Response: Thank you for your valuable comment and suggestion. Ref. [44] is the first work to explore the saturable absorption characteristics of TiS2, which is of great significance. Our experiment is a further optimization of this experiment. By controlling different cavity lengths, stable Q-switched operations can be obtained. In the mode-locked pulse operation, although the center wavelength obtained in this experiment has a blue shift for the same nanomaterial, this is our next step to explore the influence factors of the blue shift of the laser wavelength. The design of the two experiments is different. Through the ratio of different single-mode fibers, the net dispersion in the cavity is different, and the preparation method of the saturable absorber is also different, Ref. [44] used the optically deposition method, the sandwich structure was adopted in this experiment, in contrast, the saturable absorber prepared by us is easier to control and transfer, and the preparation method is simple and the cost is low. In Fig. 3(e), the curve was fitted by the function shown in the following equation:
In the article, we listed a table to compare the differences with Ref. [44].
The theoretical predictions based on a model of laser with saturable absorber as follows. As a typical 2D material, the saturable absorption mechanism can be explained by the Pauli blocking principle. Under low excitation intensity, linear absorption will occur. When the energy of the incident light is larger than the bandgap value of the TiS2, electrons distributed in the valence band can absorb the energy of the incident light and be excited into the conduction band. After that, the hot electrons cooled almost immediately and led to the formation of a hot Fermi-Dirac distribution. Under this condition, the newly created electron-hole pairs will block the originally potential interband optical transitions around the Fermi energy (-E/2) and the absorption of photons. Finally, electrons and holes recombine and reach to an equilibrium distribution state due to the intraband phonon scattering. However, under a higher excitation intensity, the concentration of photocarriers increase instantaneously, and the energy states near the edge of the conduction and valence band will be filled. The absorption will be blocked due to the fact that no two electrons can reach the same state defined by the Pauli blocking principle. Thus, specific frequency photons transmit the material without absorption. As is described, the bandgap value is of great significance in designing ultra-fast photonics devices.
The symbols and colors in Fig. 9(a) and Fig. 7(a) have been consistent, and the data in Fig. 7(a) as functions of the Pump power replaced Output power.
The English words have been modified.
Correction is marked with red colour and underline in the manuscript and listed below.
In Line 71 Page 2
In recent work, different methods have been adapted to prepare saturable absorbers, such as optically deposition, chemical vapor deposition (CVD), and deposition in tapered fibers. Saturable absorber with a sandwich structure prepared a PVA film was applied in this experiment, the saturable absorber prepared by this method is easy to control and transfer, and the preparation method is simple and the cost is low.
In Line 75 Page 2
The film-type TiS2-polyvinyl alcohol (PVA) SA used in our experiment was prepared by the method of the liquid-phase exfoliation and spin coating.
In Line 77 Page 2
The solution was placed in an ultrasonic cleaner for 6 h, few-layer TiS2 nanosheets was got.
In Line 80 Page 2
Then, 50 μL dispersion solution was spin coated on a culture dish to form TiS2-PVA film, then putting the culture dish into a drying oven at 35 ℃ for 12 h.
In Line 82 Page 2
a piece of the film with the size of ~ 1×1 mm2 was gently cut off and attached on a clean FC/PC fiber ferrule as a proposed modulator.
In Line 120 Page 4
The saturable absorption theory can be explained by the Pauli blocking principle as a typical 2D material. Linear absorption can occur under low excitation intensity. The electrons distributed in the valence band can absorb the energy of the incident light and be excited into the conduction band when the energy of the incident light is greater than the bandgap value of the TiS2. Afterwards, the hot electrons cool down almost immediately and cause the formation of a hot Fermi-Dirac distribution. In this case, the newly generated electron-hole pairs will prevent the initial potential interband optical transitions and photons absorption around the Fermi energy (-E/2). Finally, due to the internal scattering of phonons , electrons and holes recombine and reach to a state of equilibrium distribution. However, the concentration of photocarriers will increase instantaneously and the energy states near the edges of the conduction band and valence band will be filled under higher excitation intensity. Since no two electrons can reach the same state defined by the Pauli blocking principle, the absorption will be blocked. Therefore, specific frequency photons transmit the material without absorption. As is described, the bandgap value have a great significance to the design of ultra-fast photonics devices.
In Line 182 Page 7
Figure 7. (a) Variation of repetition rate and pulse duration with pump power.
In Line 185 Page 7
As sketched in Fig. 7(a), the pulse duration reduces from 7.27 to 3.42 µs and the repetition rate increases from 16.6 to 26.7 kHz with the increase of the pump power.
In Line 217 Page 9
Table 1. A Comparison of the characters of our fiber laser with Ref. [44].
αs, modulation depth; Isat, saturable intensity;
In 2018, Zhu et al reported the TiS2 used as SA for the first time, which is of great significance [44]. As shown in table 1, we compared the parameters of the laser obtained by our experiment with the parameters of their laser. They used the optically deposition method to prepare SA while the sandwich structure was adopted in our experiment, the optical modulation depth of our SA was 13.19% higher than 8.3% in their experiment. In the mode-locked and Q-switched pulse operation, although the center wavelength obtained in this experiment has a blue shift for the same nanomaterial, this is our next step to explore the influence factors of the blue shift of the laser wavelength. They can get the pulse duration of 812 fs in mode-locked pulse, and 2.36 ps pulse was obtained in our work. Higher single pulse energy was obtained in both mode-locked and Q-switched operation than Ref.[44], the highest single pulse energy can reach 67.24 nJ. This shows that the fiber laser we designed has excellent characteristics.
In Line 37 Page 1
However, its also possesses obvious disadvantages such as high cost, narrow absorption band and low damage threshold [19,20].
In Line 41 Page 2
Graphene possesses a wide absorption spectrum band due to its Dirac-like electronic band structure [22,23],
In Line 87 Page 3
the TiS2 nanosheets possess obvious layered structure.

Round 2
Reviewer 2 Report
In the new version the differences between the experiment performed in this paper and that of [44] are clearly evidenced.
In my first report I have also suggested to compare the experimental results with those based on a theoretical model. I had in mind a dynamical model which allows to calculate all the measured quantities like pulse intensity and width and repetition rate. Instead in the new version there is only a qualitative explanation of the equation written at line 110, based on Pauli's exclusion principle.
Nevertheless, I admit that my request was probably excessive and perhaps the authors will consider that issue in a next paper.